# The healthy context paradox of bullying victimization and academic adjustment among Chinese adolescents: A moderated mediation model

Yongqi Huang[ID][☯], Xiong Gan[ID][☯]*, Xin Jin, Zixu Wei, Youhan Cao, Hanzhe Ke

Department of Psychology, College of Education and Sports Sciences, Yangtze University, Jingzhou, China

☯ These authors contributed equally to this work.

* 307180052@qq.com

**Data Availability Statement:** All relevant data are within the paper and its Supporting information files.

## Abstract

Few empirical studies have specifically examined the underlying mechanisms of the "healthy context paradox" in Chinese cultural context. By constructing a moderated mediation model, the present study investigated the relationship between bullying victimization and academic adjustment, as well as the mediating effects of subjective well-being and the moderating role of classroom-level victimization. A sample of 631 adolescents ($M_{age}$ = 13.75, $SD$ = 0.74, 318 boys) were recruited from four schools in Hubei, Southern China. Results show that: (1) classroom-level victimization moderates the relationship between bullying victimization and academic adjustment. (2) Classroom-level victimization moderates the association through subjective well-being. This study confirms the healthy context paradox of bullying victimization and first reveals the mechanism of the mediating role of subjective well-being. Understanding the mechanisms that contribute to the health context paradox is crucial for developing targeted intervention strategies for individuals who experience ongoing bullying.

## 1. Introduction

Bullying is usually defined as the intentional, repeated and sustained negative behaviors of the bullied by one or more peers [1]. A latest study of 4360 junior high school students revealed that 81 (1.9%) were bullies, 647 (14.8%) were victims, and 196 (4.5%) were both bullies and victims [2]. This shows that school bullying is still widespread in our lives. It is well known that bullying in schools poses many externalizing or internalizing problems, both for the bullies and the victims, such as truancy, academic problems, depression and anxiety [3–5], and in more serious cases, self-harm or suicide [6]. And since the main task of adolescents as students is to study, academic adjustment has become a severe problem caused by bullying perpetration. There is accumulating evidence that victimized adolescents in healthy contexts had more severe adaptation problems, that is, in classrooms or schools where classroom-level victimization is low [7, 8]. Researchers have named this phenomenon the "healthy context paradox" [9, 10]. The development of this theory has attracted the attention of many researchers and

**Funding:** This research was supported by Youth project of Science and Technology Research Plan of Department of Education of Hubei Province in 2020 (Q20201306), the Social Science Fund Project of Yangtze University in 2022 (2022csz03), the Faculty Scientific Fund Project of the College of Education and Sports Sciences of Yangtze University in 2022 (2022JTB01), and the key projects of education science plan of Hubei Province in 2022: Study on the influencing factors and intervention mechanism of non-suicidal self-injurious behaviors in adolescents (2022GA030). The funders had no role in study design, data collection and analysis, decision to publish, or preparation of the manuscript.

**Competing interests:** The authors have declared that no competing interests exist.

provided an explanation and basis for its validity [11]. However, the role and mechanisms of the healthy context paradox are not fully understood, nor is there sufficient empirical evidence to demonstrate the validity and general applicability of this paradox. Therefore, the results of this study may help to explain the theory and provide empirical support for it to some extent.

## 1.1. Healthy context paradox

A "healthy" context is one where the individual's surroundings are positive, healthy, low-harm and have better external conditions for development than the average environment. Most studies use three indicators, including lower levels of bullying and victimization (the average of all students bullying and victimization levels within the class), a lower incidence of bullying and victimization, and higher levels of anti-bullying attitudes in the class [8, 12]. When victimization levels decrease in a given context, it has positive effects not only on those who avoid being victims but also on their peers at school [13]. For example, classrooms with low levels of victimization are associated with reduced social anxiety and depressive symptoms among children [14, 15]. However, a number of empirical studies have found that in this healthy context, the victims have more adjustment problems [9, 16]. This phenomenon may be the result of a mismatch between the individual and the environment [17].

As evidence for the healthy context paradox, previous research has shown that a healthy environment enhances the relationship between bullying and internalizing problems. The initial studies on this issue were conducted by Juvonen and colleagues. They discovered that the link between peer victimization and emotional distress was more pronounced in classrooms with lower levels of social disorder, which was measured as classroom levels of victimization and aggression [14]. For example, Gini et al. [16] found that bullied children and adolescents were more likely to report somatization problems in classes with lower levels of bullying. Recent longitudinal studies have also indicated that adolescents who were consistently bullied had more depressive symptoms and lower self-esteem in schools with effective anti-bullying programs and dropping bullying rates [8, 9]. In summary, prior studies suggest that victimized children are more likely to show signs of adaptation problems in classrooms with low levels of victimization compared to those with high levels. In contexts where victimization is low, these children may be less desirable as friends and more prone to negative self-evaluations [17]. Previous studies have examined the moderating role of healthy context in the relationship between experiences of bullying victimization and internalizing problems such as anxiety and depression. Nevertheless, few studies have explored the applicability of the healthy context paradox to the relationship between experiences of bullying victimization and externalizing problems such as academic adjustment. Understanding the mechanisms that contribute to the health context paradox is crucial for developing targeted intervention strategies for individuals who experience ongoing bullying.

## 1.2. Bully victimization and academic adjustment

Academic adjustment is a behavioral process where the individual, based on the needs of learning and the environment, strives to adjust themselves to achieve balance with the learning environment [18]. It is a long-term self-adjustment process that changes with the learning environment and has an impact on students' academic performance [18, 19]. Some studies have identified factors that influence academic adjustment as the learning environment, mental health and behavior [20, 21]. Children's experiences of bullying are a predictor of externalizing problems [22]. Peer victimization leaves students in a disadvantageous situation that seriously harms their mental health, leading to learning maladaptation. Furthermore, bullying victimization causes isolation from mainstream peer groups, a loss of opportunities to learn

social skills, and more deviant peers, which can contribute to more adjustment problems for victimized children [23]. There may also be a healthy context paradox between bullying victimization and adjustment problems. As the mismatch between classroom environment and individual experience can contribute to a range of adjustment problems, victimized children in a healthy context may also exhibit more academic adjustment problems.

The reason why a "healthy" classroom environment worsens the academic adjustment of victimized students may be that it hampers their interpersonal relationships, especially friendships [24]. The person-group dissimilarity model suggests that group members' attitudes towards children are influenced by the norms within the group [25]. Consequently, victims are often seen as social outcasts in classrooms with low levels of victimization and are more likely to face rejection from the mainstream peer group [26]. In classrooms where victimization is rare or nonexistent, it becomes even more challenging for victimized children to form friendships since there are few or no peers who can relate to their experiences. In this study, classroom-level victimization was chosen as an indicator of a healthy context [11, 16], to examine its moderating role between experiences of bullying victimization and academic adjustment. The classroom-level victimization is the average of the bullying victimization of all students in the class. As the victims' adjustment problems may be caused by a mismatch between class bullying and individual bullying, classroom-level victimization is more relevant to the victimized children's adjustment than other indicators (e.g., level of aggression in the class, witnessing others being bullied). Therefore, it is necessary to further explore the moderating role of classroom-level victimization between bullying and academic adjustment.

## 1.3. Mediating role of subjective well-being

Subjective well-being is a comprehensive assessment of a person's quality of life according to their own standards [27]. Subjective well-being is related to perceptions of the environment, and when students are surrounded by hostile peer relationships and bullying incidents, it can affect their psychological security and belonging. This increases the student's negative emotional experiences (e.g., anxiety and tension) and ultimately undermines their subjective well-being [28, 29]. School bullying significantly and negatively predicted junior high school students' optimism, self-esteem, positive coping and subjective well-being [30]. And optimism, self-esteem and positive coping partially mediated the effect of school bullying on subjective well-being [31]. A study of Algerian students showed that non-bullying students had higher subjective well-being than bullying victimization students and lower life satisfaction among the bullied [32]. Due to the fact that victims of bullying often lack the social skills to manage their relationships with their peers [33]. There is also a mediating effect of a positive psychological orientation between school bullying and well-being, and it gives a mechanism to mediate bullying perceptions (whether as a victim or a bully) and hence improve subjective well-being [34].

Research shows that students with high subjective well-being are more engaged in their studies so that they can achieve greater academic success [35, 36]. Salmela-Aro and Tuominen-Soini [37] found that there is a significant positive correlation between adolescent well-being and academic achievement. Moreover, considerable evidence has shown a positive correlation between subjective well-being and self-efficacy [38, 39]. Additionally, high academic self-efficacy, academic achievement, and study engagement all mean good academic adjustment [40, 41]. Students with high academic self-efficacy have a positive effect on their interpersonal and environmental adjustment, which stimulates their interest in learning and leads to greater academic adjustment. However, few studies have explored the relationship between subjective well-being and academic adjustment and the mediating role of subjective well-being

on bullying victimization and academic adjustment. Given the prior findings and theory, the present study is intended to provide a complement.

## 1.4. The present study

Previous research has found that in classes with lower average levels of bullying, victims have poorer adjustment. Few studies, however, have been conducted to investigate the impact of classroom-level victimization on the learning adjustment problems of bullying victimization and the underlying mechanisms. The present study examines whether classroom-level victimization moderates such victimization-adaptation associations and furthers the mediating role of subjective well-being between bullying victimization and academic adjustment. Hypothesis: (1) Classroom-level victimization moderates the association between bullying victimization and academic adjustment, victimized children in low classroom-level victimization are likely to exhibit more academic adjustment problems. This may be due to the fact that in low-victimization classrooms, where there is relatively little support and help among students, victims may feel more isolated and helpless, which can affect their academic performance. (2) In lower classroom-level victimization, bullying victimization will reduce children's subjective well-being and cause them to exhibit more academic adjustment problems. In a further step, we added grade, gender, parental education level and classroom size as control variables. Classroom size was indicated by the number of students in each measured class. By controlling for these variables, the study aimed to ensure that any observed effects were specifically related to the variables of interest rather than being influenced by other demographic or contextual factors. Overall, this study provides new insights into the complex interplay between classroom-level victimization, bullying victimization, subjective well-being, and academic adjustment. By understanding the role of the classroom environment and subjective well-being, educators and policymakers can develop targeted interventions and support systems to mitigate the negative consequences of bullying and enhance students' academic well-being.

## 2. Method

### 2.1. Participants and procedure

Using random cluster sampling, we recruited students from four junior middle schools in Hubei, China, in May 2022. Our survey was carried out on a class basis, and a total of 631 adolescents ranging in age from 11 to 16 years ($M_{age}$ = 13.75 years, $SD$ = 0.74, 318 boys) participated in this study. Before the formal investigation, several junior high schools were properly negotiated with for the timing of the test administration, and we obtained informed consent from the participants and their parents or legal guardians involved. Adolescents were informed that their personal information would be kept confidential, and they had the right to refuse to respond whenever they wanted. The questionnaires of the students who participated voluntarily were collected after they finished, while the remainder were not required to. The survey was carried out by well-trained psychology postgraduates during school time. During the survey, emphasis was placed on the significance of truthful responses and their contribution to the research in order to encourage truthful self-report. The present study was approved by the Research Ethics Committee of the College of Education and Sports Sciences, Yangtze University.

### 2.2. Measures

**2.2.1. Bullying victimization.** The bullying questionnaire in the Chinese version of the Olweus Bullying Victimization Questionnaire, revised by Zhang et al. [42], was used to assess

the participants' bullying victimization in school. It comprised six items (e.g., "Certain students spread some rumors about me and tried to make others dislike me"), which were rated on a 5-point scale ranging from 1 (never) to 5 (several times a week). Responses were averaged across the six items, with a higher score indicating a higher level of bullying victimization. In the present study, the Cronbach's alpha was 0.83.

**2.2.2. Subjective well-being.** We used the Index of Well-being Scale to assess adolescents' subjective well-being [43]. It consisted of nine items in total (e.g., "How satisfied or dissatisfied are you with life in general?"), which could be divided into two dimensions, including an index of general affect and life satisfaction rated on a 7-point scale. In general affect part, participants need to choose between the two ends of the emotional spectrum, with the closer the score the stronger the emotion (e.g., 1 = tiring, 7 = interesting). Responses were averaged across the nine items, with a higher score representing a higher level of subjective well-being. In the present study, the Cronbach's alpha was 0.77.

**2.2.3. Classroom-level victimization.** Referring to previous studies [11, 15, 16, 44], the mean of all students bullying scale scores in class was used as an indicator of classroom-level victimization. This means that classroom-level victimization is measured at the class level.

**2.2.4. Academic adjustment.** We used the Adaptive Behavior for Learning subscale of the Adolescent Social Adjustment Scale to measure adolescents' academic adjustment [19]. It consisted of thirty items in total (e.g., "I have my own study plan"), which could be divided into five dimensions, including learning habits, learning styles, use of learning resources, learning motivation, and learning satisfaction. Participants chose the behaviors described in the question according to their own situation, with one point for "Yes" and two points for "No". Responses were summed across all items, with a higher score meaning a higher level of academic maladjustment. In the present study, the Cronbach's alpha was 0.82.

## 2.3. Data analysis

SPSS 26.0 and PROCESS were used for processing the data. First, given that data were collected through a self-reported questionnaire, we conducted the Harman single factor method to test the shared method biases. Second, we performed descriptive statistics and bivariate correlations for the key variables. Third, we used the PROCESS to examine the mediation and moderation effects.

# 3. Results

## 3.1. Common method bias analyses

Given the possibility of common method bias in the self-report method, the common method bias was examined using the Harman single factor test procedure [45]. Results showed that there were 13 factors with a characteristic root greater than 1. The first factor of them explained a variation of 16.02%, much less than the 40% critical value. That is, the results in the present study were less influenced by the shared method biases.

## 3.2. Descriptive statistics and correlation analyses

Means, standard deviations, and bivariate associations are shown in Table 1. Particularly, bullying victimization was negatively correlated with subjective well-being ($r = -0.17$, $p < 0.001$) and positively correlated with academic adjustment ($r = 0.31$, $p < 0.001$), meaning that adolescents who experience more bullying victimization have lower subjective well-being and more academic adjustment problems. Subjective well-being was negatively associated with academic adjustment ($r = -0.22$, $p < 0.001$). Grade was negatively associated with classroom-level

**Table 1. Descriptive statistics and correlations of individual (Level 1) and classroom-level (Level 2) variables.**

|  | *M* | *SD* | 1 | 2 | 3 | 4 | 5 |
|---|---|---|---|---|---|---|---|
| Level 1 |  |  |  |  |  |  |  |
| 1 Gander | — | — | — |  |  |  |  |
| 2 Mother's education level | 2.23 | 0.88 | 0.06 | — |  |  |  |
| 3 Father's education level | 2.37 | 1.19 | 0.05 | 0.50*** | — |  |  |
| 4 Bullying victimization | 1.39 | 0.68 | 0.06 | −0.02 | 0.02 | — |  |
| 5 Subjective well-being | 3.85 | 0.73 | −0.05 | 0.04 | -0.06 | −0.17*** | — |
| 6 Academic adjustment | 1.34 | 0.21 | −0.04 | −0.06 | -0.02 | 0.31*** | −0.22*** |
| Level 2 |  |  |  |  |  |  |  |
| 1 Grade | — | — | — |  |  |  |  |
| 2 Classroom size | 30.56 | 4.18 | 0.33*** | — |  |  |  |
| 3 Classroom-level victimization | 1.39 | 0.30 | −0.08* | 0.07 | — |  |  |

Note.

*$p < .05$,

**$p < .01$,

***$p < .001$.

victimization ($r = −0.08$, $p < 0.05$), indicating that students with high subjective well-being have fewer academic adjustment problems.

### 3.3. Mediation effect and moderation effect analyses

Frist, the results of the linear regression test showed that, without considering the effect of other variables, bullying victimization could predict academic adjustment directly and positively ($\beta = 0.31$, $t = 8.02$, $p < 0.001$). Second, model 4 in PROCESS was used to test the moderated mediation model. After controlling gender, grade, parental education level and classroom size, bullying victimization could negatively predict subjective well-being ($\beta = −0.16$, $t = −4.12$, $p < 0.001$) and positively predict academic adjustment ($\beta = 0.28$, $t = 7.24$, $p < 0.001$); subjective well-being could negatively predict academic adjustment ($\beta = −0.17$, $t = −4.50$, $p < 0.001$). Bootstrapping analyses were used to test the relationship between each path and if the 95% confidence interval did not include 0, then the indirect effect was significant. The result revealed that subjective well-being mediated the relationship between bullying victimization and academic adjustment ($\beta = 0.03$, *95% CI* = [0.01, 0.05]; see Fig 1), indicating that bullying victimization not only directly predicts academic adjustment but also predicts academic adjustment through the mediating effect of subjective well-being.

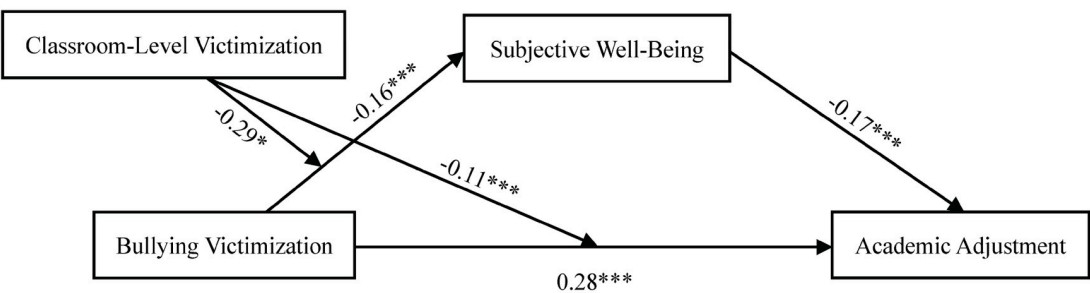

**Fig 1. Test of the moderated mediation model.** *$p < 0.05$, ***$p < 0.001$.

Third, model 8 in PROCESS was used to assess the moderating effect of classroom-level victimization on the relationships between bullying victimization and academic adjustment, as well as bullying victimization and subjective well-being. Controlling gander, grade, parental education level and classroom size, the interaction term (bullying victimization × classroom-level victimization) could negatively predict subjective well-being and academic adjustment ($\beta$ = −0.29, $t$ = −2.06, $p < 0.05$; $\beta$ = −0.11, $t$ = −3.20, $p < 0.001$; see Fig 1), indicating that classroom-level victimization moderated the direct pathway from bullying victimization to academic adjustment and the predictive effect of bullying victimization on subjective well-being. Then, according to the score of classroom-level victimization, adolescents were divided into two groups. The high/low classroom-level victimization group included adolescents whose score was one standard deviation (SD) above/below the average. And we drew two simple slope pictures (see Figs 2 and 3). As shown in Fig 2, in classes with a low level of victimization (M−1SD), bullying victimization had a positive predictive effect on academic adjustment, *simple slope* = 0.14, $t$ = 5.81, $p < 0.001$. For classes with a high level of victimization (M+1SD), although bullying victimization also has a positive predictive effect on academic adjustment, it is minor, *simple slope* = 0.08, $t$ = 6.94, $p < 0.001$. This suggests that the predictive effect of bullying victimization on academic adjustment increases with classroom-level victimization lowering. As can be seen in Fig 3, when adolescents were in classes with a low level of victimization (M−1SD), bullying victimization had a negative predictive effect on subjective well-being, *simple slope* = −0.19, $t$ = −4.05, $p < 0.001$. That indicated that in classrooms with lower levels of victimization, the effect of bullying victimization on subjective well-being became bigger.

## 4. Discussion

By constructing a moderated mediation model, the present study investigated the relationship between bullying victimization and adolescents' academic adjustment. It was found that classroom-level victimization moderated the association between bullying victimization and academic adjustment, with stronger associations between children's bullying victimization and academic adjustment problems in classes with low classroom-level victimization. Moreover, lower classroom-level victimization decreases the victimized child's subjective well-being, which then predicts academic adjustment problems. These findings provide further evidence of the health context paradox between individual bullying victimization and academic adjustment and the first evidence of the mediating role of subjective well-being.

The results of this study are consistent with our hypothesis and the previous studies [46, 47], and we found that the association between bullying victimization and academic adjustment was stronger in healthy contexts (i.e., low classroom-level bullying victimization levels). As shown in Fig 2, the simple slope picture shows that frequently victimized students in healthy contexts with lower classroom-level victimization are likely to exhibit more academic adjustment problems. This result also provides further support for the healthy context paradox of bullying victimization and academic adjustment. This may be due to a mismatch between individual experiences of bullying victimization and the low levels of bullying victimization in the classroom, causing the victimized children to show more adjustment problems [26]. In healthy contexts, where the number of bullied individuals is low, the bullied adolescents lack peers who have similar experiences for social comparisons. They can only make upward social comparisons with those who are not victims, which can lead to more negative emotional problems [48, 49] and thus maladaptive academic problems.

In view of the individual level, according to person-environment fit theory, stress increases when individuals perceive a mismatch between their environment and themselves

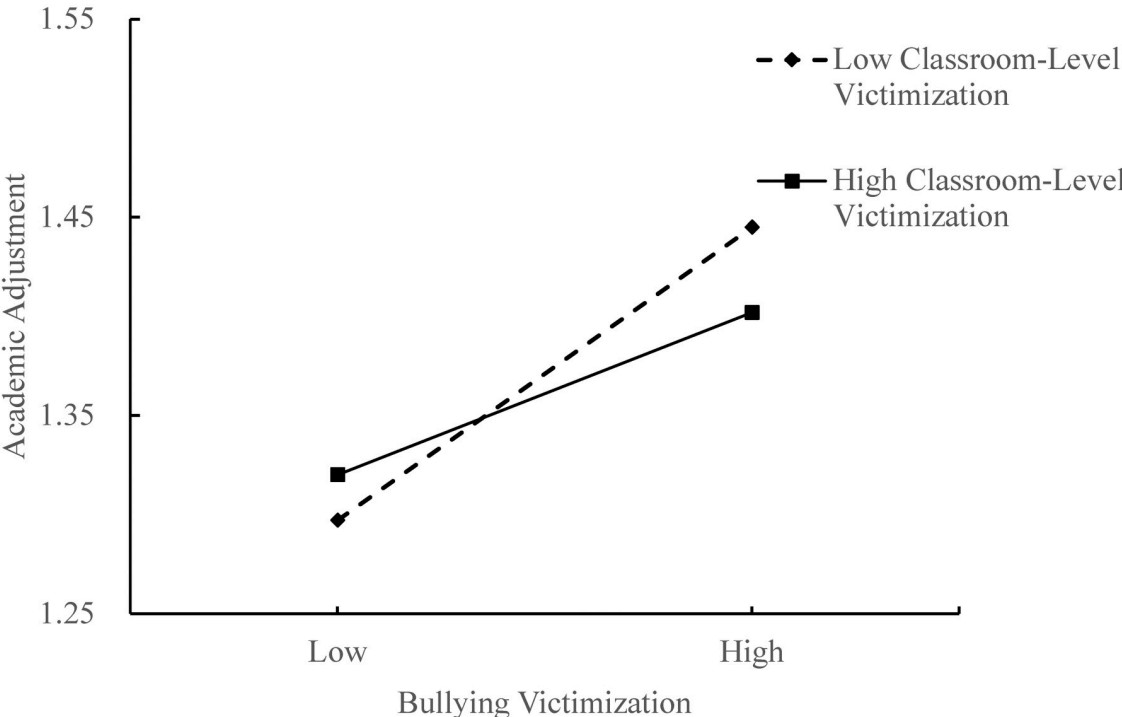

**Fig 2. Moderating effect of classroom-level victimization in the relationship between bullying victimization and academic adjustment.**

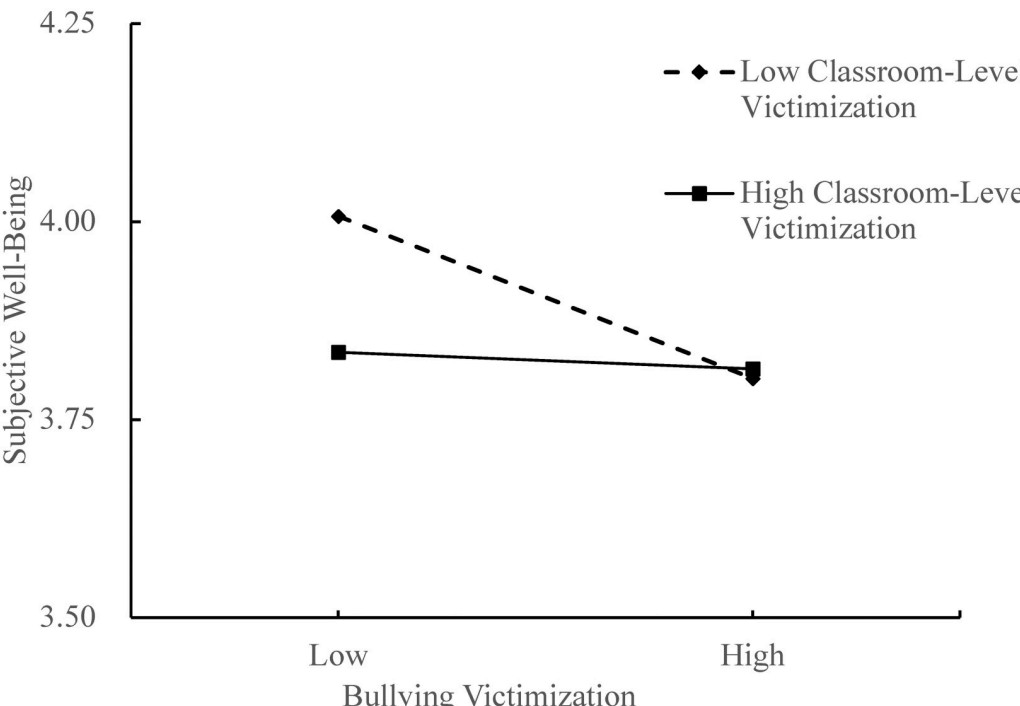

**Fig 3. Moderating effect of classroom-level victimization in the relationship between bullying victimization and subjective well-being.**

[50]. In a healthy classroom environment, bullying contributes to sensitivity to exclusion and rejection, generating a great contrast between the surroundings and the attitudes of others, which produces stress and a reduced sense of belonging at school [51]. Students who do not feel belonging and security will develop a variety of academic problems [52]. In view of the environment level, classroom-level victimization reflects the environment level of a class group. If the classroom-level victimization is high, then the class is considered to be an unhealthy environment; if the classroom-level victimization is low, then the class is considered to be a healthy environment. According to social identity theory, people have more apparent perceptions of similar characteristics within a group and more apparent perceptions of differences between dissimilar objects [53]. Therefore, when an individual is bullied in a low-victimization classroom environment, he or she becomes a 'minority' and an 'anomaly' in the class. In Wright's theory of social status, people do not understand why they are bullied, and exclusion and rejection of them leave the victimized adolescents without safety and belonging, which causes academic difficulties [25]. Furthermore, because there is little or no bullying in a healthy classroom environment, it is difficult for the bullied student to find classmates who have gone through similar experiences to support and comfort each other, and they must bear and disperse their pain alone. But the immature minds of teenagers are not yet capable of supporting them to overcome their negative emotions, and this leads to more serious emotional problems. Conversely, the negative effects of a relatively poor classroom environment are less likely to be experienced. In classrooms where victimization is more common, students are accustomed to such behavior and even being bullied, thus bullied children are not seen as different. The bullied are more likely to receive consolation from their peers; therefore, being bullied in a poor environment might alleviate stress and emotional issues.

To unravel the health context paradox of bullying victimization and academic adjustment, we examined the mediating role of subjective well-being. It was found that bullying victimization indirectly predicted academic adjustment by affecting subjective well-being. Bullying victimization causes a decrease in subjective well-being, which then leads to an increase in academic adjustment problems, with the mediating effect being more prominent in healthy environments. Additionally, this present finding is consistent with previous studies [54–56]. This study also found that adolescents' subjective well-being predicted their academic adjustment problems. Subjective well-being has a direct impact on students' emotional experiences during the learning process, when subjective well-being is high, they tend to have positive emotional experiences. This positive emotion will enhance their academic adjustment, for instance, by increasing their recognition of school activities [57]. Positive emotions have a positive effect on students' ability to pay attention and remember things, as well as on their ability to deal with learning stress and problems [58]. They are also linked to higher motivation, better independent learning, and better test scores [59]. In general, bullying victimization has a direct impact on academic adjustment but also has a partially mediated effect on subjective well-being. This finding provided the first insight into the mechanisms underlying the healthy context paradox of bullying and academic adjustment and shed light on how classroom-level victimization affects the academic adjustment of victimized children. Specifically, this study found that when victimized youth are exposed to low classroom-level victimization, it negatively affects their academic performance due to a decrease in subjective well-being. These findings highlight why seemingly positive environments can still have a detrimental impact on victimized children's academic adjustment. It emphasizes the importance of not only implementing effective antibullying programs but also providing additional support and assistance to specific children who are at risk of continued victimization.

## 5. Implications, limitations and future directions

The results of this study extend and deepen the understanding of existing research on the relationship between the classroom environment and the academic adjustment of victims of bullying. Actually, this study is the first to demonstrate that a healthy classroom environment can contribute to the low subjective well-being and academic maladjustment of victimized children. We found that the subjective well-being of the bullied children was lower in healthy contexts. Another contribution of this research is examining a sample of Chinese youth participants. Our findings align with previous research conducted in Western countries, which has shown that classroom-level victimization plays a moderating role in the relationship between victimization and adjustment [8, 9, 16]. Additionally, we have identified a mediating mechanism that explains this phenomenon. While our hypotheses were originally derived from well-established theories and empirical studies in Western culture, it is worth considering that these mechanisms could potentially apply to other cultures as well. However, it is crucial to acknowledge that due to the emphasis on social harmony and interdependence in Chinese culture, victimized children within more supportive environments may face heightened rejection and develop negative social self-perceptions. Future research may be able to focus on adolescents' interpersonal relationships and cognitive processes to explore other mediating roles in more depth. We believe that our results are instructive and deserve the attention of our interventionists. In prevention and intervention, we aim to reduce the incidence of bullying and the number of victimized students, but in the process, adolescents who continue to experience bullying may exhibit more emotional and academic problems. Therefore, while improving the classroom environment, teachers need to take a proactive approach to the psychology and academic performance of bullied students. More specifically, it is crucial to enhance and modify anti-bullying programs, especially in challenging situations where victimization cannot be immediately stopped [24]. Anti-bullying initiatives should not only focus on preventing incidents and reducing the number of victims overall but also on identifying specific victims and recognizing their status within the classroom. Schools should have policies that facilitate reporting of victimization, effective handling of cases, and monitoring to ensure successful adult intervention. Moreover, during the intervention, teachers should improve students' subjective well-being. Specifically, parents and teachers should give prompt care and attention to students after bullying incidents to reduce their negative emotions. Teachers can also design positive and interactive games and activities to help students develop good peer beliefs and gain friendship support, encouraging peers to support victimized children in improving their situation [60]. Additionally, teachers can provide victimized children with information about others who are also experiencing victimization. This helps the victims understand that they are not alone, and that the situation is not their fault.

The present research also has some limitations that should be noted. First, this study used a cross-sectional design, revealing only the concurrent associations between bullying victimization, subjective well-being, and adolescent academic adjustment. These three may have a complicated two-way relationship [61–63]. Therefore, longitudinal research is necessary to examine whether there is an interaction between bullying victimization, subjective well-being, academic adjustment and classroom-level victimization. Secondly, this study focused on only one possible indicator of classroom environments (i.e., low classroom-level victimization) that might have a substantial influence in determining the variance in the association between victimization and academic adjustment. Nonetheless, it would be intriguing to investigate the moderating effect of additional environmental factors. For example, researchers have found that classroom climate is strongly associated with bullying behavior among primary school students, and the most harmonious classroom atmosphere is perceived by the uninvolved [64].

Examining these various contextual factors would give a wider view of how classroom environments impact the academic adjustment of victims. Third, a limitation of this study in using data analysis methods is that we did not eventually use hierarchical linear models. Considering the problem of estimation bias or standard error that may arise from traditional regression models, in the future we may consider constructing multi-level models with larger sample sizes and better organized data structures. Finally, data was collected only through self-reported measures. Self-reporting may be subject to more bias (e.g., socially desirable responses), and participants may report less bullying victimization and academic adjustment problems. Reports from multiple informants (e.g., parents, teachers, and peers) should be considered in future research.

## 6. Conclusion

The findings of this study support the existence of the healthy context paradox and its mechanisms in the relationship between bullying victimization and academic adjustment. Consistent with previous research, our study revealed that victimized children are more prone to academic maladjustment when they experience low levels of victimization in their classrooms compared to high levels. Specifically, it includes two aspects: (1) classroom-level victimization moderates the relationship between bullying victimization and academic adjustment. In classrooms with low levels of victimization, victimized children exhibit more academic adjustment problems; (2) the classroom-level victimization moderates the associations through subjective well-being, a lower classroom-level victimization increases academic adjustment problems by reducing the subjective well-being of the victims.

## Supporting information

**S1 Dataset. Dataset used for analyses in present study.**
(SAV)

## Author Contributions

**Conceptualization:** Xiong Gan.

**Data curation:** Yongqi Huang, Xin Jin, Zixu Wei, Youhan Cao.

**Funding acquisition:** Xiong Gan.

**Writing – original draft:** Yongqi Huang.

**Writing – review & editing:** Xiong Gan, Hanzhe Ke.

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
