## [Decision Letter · Decision Letter 0]

29 Jun 2023

PONE-D-23-06364The Healthy Context Paradox of Adolescent Bullying Victimization and Academic Adjustment: A Moderated Mediation ModelPLOS ONE

Dear Dr. Gan,

Thank you for submitting your manuscript to PLOS ONE. After careful consideration, we feel that it has merit but does not fully meet PLOS ONE’s publication criteria as it currently stands. Therefore, we invite you to submit a revised version of the manuscript that addresses the points raised during the review process.

We look forward to receiving your revised manuscript.

Kind regards,

Gianpiero Greco

Academic Editor

PLOS ONE

Journal Requirements:

This research was supported by Youth project of Science and Technology Research Plan of Department of Education of Hubei Province in 2020 (Q20201306), the Social Science Fund Project of Yangtze University in 2022 (2022), the Faculty Scientific Fund Project of the College of Education and Sports Sciences of Yangtze University in 2022 (2022JTB01), and the key projects of education science plan of Hubei Province in 2022: Study on the influencing factors and intervention mechanism of non-suicidal self-injurious behaviors in adolescents (2022GA030). 

Additional Editor Comments:

Dear authors,

please reply point by point to the reviewers' comments.

Reviewers' comments:

Reviewer's Responses to Questions

**Comments to the Author**

1. Is the manuscript technically sound, and do the data support the conclusions?

Reviewer #1: Partly

Reviewer #2: No

2. Has the statistical analysis been performed appropriately and rigorously? 

Reviewer #1: Yes

Reviewer #2: No

3. Have the authors made all data underlying the findings in their manuscript fully available?

Reviewer #1: Yes

Reviewer #2: Yes

4. Is the manuscript presented in an intelligible fashion and written in standard English?

Reviewer #1: Yes

Reviewer #2: No

5. Review Comments to the Author

Reviewer #1: Comments to authors:

I am pleased to receive the invitation to review this paper entitled " The Healthy Context Paradox of Adolescent Bullying Victimization and Academic Adjustment: A Moderated Mediation Model", which explores the relationship between bullying victimization and academic adjustment, as well as the mediating effects of subjective well-being and the moderating role of classroom-level victimization. My comments are as follows:

Introduction

(1) It would help if the author provides more theoretical and empirical base for the debate of “Healthy Context Paradox”.

Methods and results

(2) The authors should give more detailed information about their random cluster sampling process.

(3) The measures used in the study suffer from several drawbacks. The most important one is that it is unclear whether the classroom-level victimization variable is a class level measure or an individual level measure. It must be clarified at which level it has been measured and included in the model, and the modeling must adjust accordingly. In my view, the study could benefit from re-analysis using more appropriate and advanced statistical methods. The authors could have considered using multi-level models, also known as hierarchical linear models, to account for the potential issue of biased estimates or standard errors that can arise from traditional regression models.

(4) Bootstrap analysis could be helpful to test the mediating effect.

Discussion

(5) Please give more specific practical implications both on policy level and management level.

(6) In addition, there are some grammar errors in your manuscript. Please check it carefully.

Reviewer #2: Thank you for the opportunity to review the manuscript. Certainly, the topic is interesting, relevant and fits the contents of the magazine. I believe it has good potential and great practical relevance in the field of education. The origin of the students should be included in the title. The abstract defines conceptually different parts of the study and summarizing the content and objectives of the work, contains the required number of words and at least five keywords. Keywords refer to the most important variables or topics of the study. The justification explained in the manuscript is shallow, some references should be updated, and secondary citations are used. Occasionally, the writing is somewhat confusing (the information should be better organized, some ideas should be lightened, and the discourse should be coherent). The sample selection procedure should be better explained. The research hypotheses should be explained after stating the objectives. You should better explain the data analysis section and justify the statistics used. I consider that the practical relevance of the results found should be explained in greater depth, not simply describing them, but explaining their practical scope and justifying them according to previous empirical evidence. Sometimes a literal description of the results is made without a discussion of them. It would be important to include a section after the limitations that includes the main practical implications, in order to better organize the information of the work. I believe that it would also be interesting to include other variables that may be mediating the relationship between the variables. It would be interesting to explain the following statement in the text (page 19, lines 3-5): "This finding provided the first insight into the mechanisms underlying the healthy context paradox of bullying and academic adjustment, and shed light on how the classroom-level victimization affects the academic adjustment of victimized children”. On a formal level, the manuscript complies with the requirements of the Journal and references are written in accordance with the regulations of the Journal. It should indicate the exact location of tables and figures in the manuscript. You should revise some English expressions and phrases to make the text more coherent. The work is ambitious, and the results confirm most of the hypotheses and the relevance and potential of the work is therefore recognized, but this Reviewer considers that several changes are needed to the manuscript is publishable. In this sense, it should better explain the novelty and relevance of the work considering the previous empirical evidence and should better describe the practical implications, and the principal limitations. It should describe the discussion and conclusions of the work better and, above all, update the manuscript references (most should be from the last 5 years). Finally, I wish the Authors the best in continuing this line of research.

Best wishes for Authors.

6. PLOS authors have the option to publish the peer review history of their article (what does this mean?). If published, this will include your full peer review and any attached files.

Reviewer #1: No

Reviewer #2: **Yes: **David Aparisi

---

## [Author Response · Author response to Decision Letter 0]

13 Jul 2023

Dear Editor and Reviewers:

Thank you for your letter and for the reviewers’ comments concerning our manuscript entitled “The Healthy Context Paradox of Bullying Victimization and Academic Adjustment Among Chinese Adolescents: A Moderated Mediation Model” [PONE-D-23-06364]. We are pleased to know that our work was rated as potentially acceptable for publication, subject to adequate revision. These comments are all valuable and very helpful for revising and refining our paper, as well as providing important guidance for our research. Based on your suggestion and request, we have carefully studied these comments and made corrections, which we hope will meet with your approval. We would also like to thank you for allowing us to resubmit a revised version of the manuscript. Revised sections are marked in red in the paper. Responses to the reviewer’s comments are as flowing:

Changes regarding financial disclosure:

The updated statement on financial disclosure is included in the cover letter, as detailed in the section marked in red.

Response to Journal Requirements:

The author’s answer: In the revised version, we carefully examined the manuscript and made changes to the parts that did not meet the norms. First, for different levels of titles, we changed the format accordingly. See lines 13, 26, 41, 68, 95, 118, 139, 140, 152, 153, 159, 166, 170, 178, 184, 185, 190, 200, 235, 296, 344, 354. Second, we added the title to the manuscript and made sure it appeared directly after the paragraph in which they were first cited. See lines 212, 231-234. Third, we checked and revised the format of all references to ensure they were consistent and met the requirements of the journal. Finally, we have included the captions of the supporting information files at the end of the manuscript. See lines 521-522.

The author’s answer: Our study was approved by the Research Ethics Committee of the Psychology, College of Education and Sports Sciences, Yangtze University. During the study, we always maintain the important principles of voluntary participation and informed consent. Since the participants in the study were minors, we were informed in advance in order to obtain the consent of their parents or guardians. During the study, their informed consent was reflected by the return of the questionnaire. If either parent or the student does not agree to participate in this survey, the student does not be required to complete the questionnaire if he or she verbally states so. In the participants section, the specific details of obtaining informed consent have been added accordingly. See lines 144-147.

The author’s answer: Thanks for pointing this out. We submitted again with the correct version.

4. We note that you have provided funding information that is not currently declared in your Funding Statement. However, funding information should not appear in the Acknowledgments section or other areas of your manuscript. We will only publish funding information present in the Funding Statement section of the online submission form. Please remove any funding-related text from the manuscript and let us know how you would like to update your Funding Statement. Currently, your Funding Statement reads as follows: The funders had no role in study design, data collection and analysis, decision to publish, or preparation of the manuscript. Please include your amended statements within your cover letter; we will change the online submission form on your behalf.

The author’s answer: We removed this information from the resubmitted manuscript. Our Funding Statement reads as follows:

Xiong Gan received fund. This research was supported by Youth project of Science and Technology Research Plan of Department of Education of Hubei Province in 2020 (Q20201306), the Social Science Fund Project of Yangtze University in 2022 (2022csz03), the Faculty Scientific Fund Project of the College of Education and Sports Sciences of Yangtze University in 2022 (2022JTB01), and the key projects of education science plan of Hubei Province in 2022: Study on the influencing factors and intervention mechanism of non-suicidal self-injurious behaviors in adolescents (2022GA030).

Response to Reviewer #1:

I am pleased to receive the invitation to review this paper entitled " The Healthy Context Paradox of Adolescent Bullying Victimization and Academic Adjustment: A Moderated Mediation Model", which explores the relationship between bullying victimization and academic adjustment, as well as the mediating effects of subjective well-being and the moderating role of classroom-level victimization. My comments are as follows:

(1) It would help if the author provides more theoretical and empirical base for the debate of “Healthy Context Paradox”.

The author’s answer: We have elaborated more on the original manuscript. See lines 46-48, 52-55, 59-61, 81-87. 

(2) The authors should give more detailed information about their random cluster sampling process.

The author’s answer: Thanks for your suggestion. We have described this in more detail. See lines140-142.

(3) The measures used in the study suffer from several drawbacks. The most important one is that it is unclear whether the classroom-level victimization variable is a class level measure or an individual level measure. It must be clarified at which level it has been measured and included in the model, and the modeling must adjust accordingly. In my view, the study could benefit from re-analysis using more appropriate and advanced statistical methods. The authors could have considered using multi-level models, also known as hierarchical linear models, to account for the potential issue of biased estimates or standard errors that can arise from traditional regression models.

The author’s answer: Thank you very much for your review and valuable suggestions on our research measures. The measurement method you mentioned has some drawbacks, we fully agree with your point of view and have made corresponding explanations in the revised draft. See lines 168-169, 337-340. In our preliminary analysis, we attempted to use hierarchical linear models to construct model and analyze data. However, we found that the Intra-Class Correlation (ICC) indicator did not meet requirements, indicating lower than expected within-group correlation which affected our interpretation and modeling of victim variables at the classroom level. In addition, according to Peugh (2010), when the design effect of the sample is greater than 2, it is necessary to analyze the data using a multilevel model to avoid statistical bias. For some reason, the design effect for one of our variables did not reach 2, so on balance we did not use a multilevel linear model. It is possible that the data structure or the sample size caused these problems, so we recognize that this is a limitation of our statistical approach, and in the future research, we will consider collecting larger samples and ensuring good data structure to construct more appropriate models.

Peugh, J. L. (2010). A practical guide to multilevel modeling. Journal of School Psychology, 48(1), 85–112. https://doi.org/10.1016/j.jsp.2009.09.002

(4) Bootstrap analysis could be helpful to test the mediating effect.

The author’s answer: Thank you for your suggestion. We have used bootstrap analysis in the process of analyzing the mediation effect. It may have been due to our oversight that we did not explicitly state it in the manuscript. We have now improved this part. See lines 206-211.

(5) Please give more specific practical implications both on policy level and management level.

The author’s answer: Thanks for your advice. We give additional and specific practical implications at both levels. See lines 314-319, 322-325.

(6) In addition, there are some grammar errors in your manuscript. Please check it carefully.

The author’s answer: Thanks for your careful review, we have checked and corrected it.

Response to Reviewer #2: 

Thank you for the opportunity to review the manuscript. Certainly, the topic is interesting, relevant and fits the contents of the magazine. I believe it has good potential and great practical relevance in the field of education. 

1. The origin of the students should be included in the title. 

The author’s answer: Thank you for your advice. We have made a change to this. See lines 1-2.

2. The abstract defines conceptually different parts of the study and summarizing the content and objectives of the work, contains the required number of words and at least five keywords. Keywords refer to the most important variables or topics of the study. 

The author’s answer: Thanks for your suggestion. We refined the abstract and keywords. See lines 14-25. 

3. The justification explained in the manuscript is shallow, some references should be updated, and secondary citations are used. 

The author’s answer: We provide additional explanations of the healthy environment paradox and update some of the references. See lines 46-48, 52-55, 59-61, 81-87.

4. Occasionally, the writing is somewhat confusing (the information should be better organized, some ideas should be lightened, and the discourse should be coherent). 

The author’s answer: Thank you for your suggestions, we have scrutinized our manuscript and made some changes.

5. The sample selection procedure should be better explained. 

The author’s answer: We made further clarifications. See lines 140-142, 144-147.

6. The research hypotheses should be explained after stating the objectives. 

The author’s answer: We have explained accordingly in the research hypotheses section. See lines 125-127, 130-136.

7. You should better explain the data analysis section and justify the statistics used. I consider that the practical relevance of the results found should be explained in greater depth, not simply describing them, but explaining their practical scope and justifying them according to previous empirical evidence. Sometimes a literal description of the results is made without a discussion of them. 

The author’s answer: We have revised the relevant sections to provide a more in-depth analysis and interpretation of the findings. See lines 193-194, 196-197, 206-211, 244, 301-303.

8. It would be important to include a section after the limitations that includes the main practical implications, in order to better organize the information of the work. 

The author’s answer: Thanks for your advice. We have added some information about this. See lines 298-311, 314-319, 322-325.

9. I believe that it would also be interesting to include other variables that may be mediating the relationship between the variables. 

The author’s answer: Thanks for your advice. We have provided some clarification on this. See lines 309-310.

10. It would be interesting to explain the following statement in the text (page 19, lines 3-5): "This finding provided the first insight into the mechanisms underlying the healthy context paradox of bullying and academic adjustment, and shed light on how the classroom-level victimization affects the academic adjustment of victimized children”. 

The author’s answer: Thanks for your review. We have explained this further. See lines 290-295.

11. On a formal level, the manuscript complies with the requirements of the Journal and references are written in accordance with the regulations of the Journal. It should indicate the exact location of tables and figures in the manuscript. 

The author’s answer: Thanks for the heads up. We've added the headings where the figure should be in. See lines 212, 231-234.

12. You should revise some English expressions and phrases to make the text more coherent. 

The author’s answer: Thanks for the mention. We've double-checked the entire article and made changes accordingly.

13. The work is ambitious, and the results confirm most of the hypotheses and the relevance and potential of the work is therefore recognized, but this Reviewer considers that several changes are needed to the manuscript is publishable. In this sense, it should better explain the novelty and relevance of the work considering the previous empirical evidence and should better describe the practical implications, and the principal limitations. 

The author’s answer: Thank you for taking the time to review our manuscript. We appreciate your valuable feedback and constructive comments. We have carefully considered your suggestions and have made the necessary revisions to address the concerns raised. See lines 298-311, 314-319, 322-325, 337-340.

14. It should describe the discussion and conclusions of the work better and, above all, update the manuscript references (most should be from the last 5 years). Finally, I wish the Authors the best in continuing this line of research.

The author’s answer: We sincerely appreciate your feedback and suggestions for improving the discussion and conclusions of our work. See lines 290-295, 345-352. We have made improvements to optimize the discussion and conclusion sections of our manuscript and replaced outdated references with more current and relevant sources.

In short, our original manuscript is relatively immature. Your suggestions have helped us clarify the variable relationships and illuminate the significance of the study. Also, your suggestions for certain misrepresentations were very timely. Based on your questions, we have carefully examined the entire text and added appropriate additions to the text where it is unclear, in order to improve it. These descriptions may not be what you asked for, but based on your suggestions, we think it is necessary after carefully combing through the article. We are grateful and will pay more attention in our future research. Thank you!

Thanks to the reviewer for their approval and suggestions, which helped us to improve the article, especially the description and explanation of the relationship between the variables. We tried our best to improve the manuscript and made some changes in the manuscript. These changes will not influence the framework of the paper. We appreciate for Editors and Reviewers warm work earnestly and hope that the correction will meet with approval.

Once again, thank you very much for your comments and suggestions.

Yours sincerely,

Xiong Gan

July 12, 2023

Department of Psychology, College of Education and Sports Sciences, Yangtze

University, Hubei, Jingzhou 434023, China

E-mail: 307180052@qq.com

---

## [Decision Letter · Decision Letter 1]

9 Aug 2023

The Healthy Context Paradox of Bullying Victimization and Academic Adjustment Among Chinese Adolescents: A Moderated Mediation Model

PONE-D-23-06364R1

Dear Dr. Gan,

We’re pleased to inform you that your manuscript has been judged scientifically suitable for publication and will be formally accepted for publication once it meets all outstanding technical requirements.

Kind regards,

Gianpiero Greco

Academic Editor

PLOS ONE

Additional Editor Comments (optional):

Reviewers' comments:

Reviewer's Responses to Questions

**Comments to the Author**

1. If the authors have adequately addressed your comments raised in a previous round of review and you feel that this manuscript is now acceptable for publication, you may indicate that here to bypass the “Comments to the Author” section, enter your conflict of interest statement in the “Confidential to Editor” section, and submit your "Accept" recommendation.

Reviewer #1: All comments have been addressed

Reviewer #2: All comments have been addressed

2. Is the manuscript technically sound, and do the data support the conclusions?

Reviewer #1: Yes

Reviewer #2: Yes

3. Has the statistical analysis been performed appropriately and rigorously? 

Reviewer #1: Yes

Reviewer #2: Yes

4. Have the authors made all data underlying the findings in their manuscript fully available?

Reviewer #1: Yes

Reviewer #2: Yes

5. Is the manuscript presented in an intelligible fashion and written in standard English?

Reviewer #1: Yes

Reviewer #2: Yes

6. Review Comments to the Author

Reviewer #1: (No Response)

Reviewer #2: Thank you for the opportunity to review the manuscript again.

Overall, the writing is clear, the goals are well described, well-considered introduction and the results properly made and presented. Therefore, the manuscript brings significant knowledge of the scientific literature so and still covers existing gaps in the field of Education. Therefore, my assessment is positive for the publication of this work, with a new suggestion.

Firstly, I would like to thank the efforts by the authors of the manuscript and congratulate them on the work. I recognize that they have considered almost all considerations of the Reviewers. Clearly, all the comments from Reviewers have contributed to a better quality of the manuscript. I have checked in the revised manuscript are corrected the most of errors found by the reviewers, both formally and content.

Secondly, I have verified that the information is presented in a clear and organized way in subtitles. It assumes good work with great potential.

Thirdly, in the Discussion section appears practical and educational implications and future directions correctly described. I have found the manuscript show a paragraph of study limitations. However, the conclusions section is somewhat concise. It is precisely a section to highlight the main practical implications of the study.

Fourthly, I have found that all the references are correctly written. The references are quite current, and all references comply with the Journal style.

Fifthly, I have verified that the format of the figures complies with the regulations of the Journal.

Finally, considering the changes made to the manuscript by the authors and the new suggestion, I consider that the manuscript can continue with the review process, considering the opinion and suggestions of other Reviewers.

Best wishes for Authors.

7. PLOS authors have the option to publish the peer review history of their article (what does this mean?). If published, this will include your full peer review and any attached files.

Reviewer #1: No

Reviewer #2: No

---

## [Editor Report · Acceptance letter]

11 Aug 2023

PONE-D-23-06364R1 

The Healthy Context Paradox of Bullying Victimization and Academic Adjustment Among Chinese Adolescents: A Moderated Mediation Model 

Dear Dr. Gan:

I'm pleased to inform you that your manuscript has been deemed suitable for publication in PLOS ONE. Congratulations! Your manuscript is now with our production department. 

Kind regards, 

on behalf of

Dr. Gianpiero Greco 

Academic Editor

PLOS ONE